# Effects of Drug Physicochemical Properties on In-Situ Forming Implant Polymer Degradation and Drug Release Kinetics

**DOI:** 10.3390/pharmaceutics14061188

**Published:** 2022-06-01

**Authors:** Jordan B. Joiner, Alka Prasher, Isabella C. Young, Jessie Kim, Roopali Shrivastava, Panita Maturavongsadit, Soumya Rahima Benhabbour

**Affiliations:** 1Division of Pharmacoengineering and Molecular Pharmaceutics, Eshelman School of Pharmacy, University of North Carolina at Chapel Hill, Chapel Hill, NC 27599, USA; jbjoiner@unc.edu (J.B.J.); iyoung4@live.unc.edu (I.C.Y.); panita@med.unc.edu (P.M.); 2Joint Department of Biomedical Engineering, University of North Carolina and North Carolina State University, Chapel Hill, NC 27599, USA; alkaprasher@gmail.com (A.P.); roopalis@email.unc.edu (R.S.); 3Eshelman School of Pharmacy, University of North Carolina at Chapel Hill, Chapel Hill, NC 27599, USA; jessiek@alumni.unc.edu

**Keywords:** long-acting, in-situ, injectable, biodegradable implants, poly(lactic-*co*-glycolic acid), controlled drug release, drug delivery, drug physicochemical properties

## Abstract

In-situ forming implants (ISFIs) represent a simple, tunable, and biodegradable polymer-based platform for long-acting drug delivery. However, drugs with different physicochemical properties and physical states in the polymer-solvent system exhibit different drug release kinetics. Although a few limited studies have been performed attempting to elucidate these effects, a large, systematic study has not been performed until now. The purpose of this study was to characterize the in vitro drug release of 12 different small molecule drugs with differing logP and pKa values from ISFIs. Drug release was compared with polymer degradation as measured by lactic acid (LA) release and change in poly(DL-lactide-*co*-glycolide) (PLGA) molecular weight (MW) measured by size exclusion chromatography with multi-angle laser light scattering (SEC-MALS). Drug physical state and morphology were also measured using differential scanning calorimetry (DSC) and scanning electron microscopy (SEM). Together, these results demonstrated that hydrophilic drugs have higher burst release at 24 h (22.8–68.4%) and complete drug release within 60 days, while hydrophobic drugs have lower burst release at 24 h (1.8–18.9%) and can sustain drug release over 60–285 days. Overall, drug logP and drug physical state in the polymer–solvent system are the most important factors when predicting the drug release rate in an ISFI for small-molecule drugs. Hydrophilic drugs exhibit high initial burst and less sustained release due to their miscibility with the aqueous phase, while hydrophobic drugs have lower initial burst and more sustained release due to their affinity for the hydrophobic PLGA. Additionally, while hydrophilic drugs seem to accelerate the degradation of PLGA, hydrophobic drugs on the other hand seem to slow down the PLGA degradation process compared with placebo ISFIs. Furthermore, drugs that were in a crystalline state within the ISFI drugs exhibited more sustained release compared with amorphous drugs.

## 1. Introduction

In-situ forming implants (ISFIs) are sustained drug delivery systems consisting of a biocompatible, water-miscible solvent such as *N*-methyl-2-pyrrolidone (NMP) or dimethyl sulfoxide (DMSO) and a biodegradable polymer, most commonly a polyester such as PLGA [1,2,3,4]. The water-miscible solvent is used to dissolve both the polymer and drug to form a liquid formulation that can be injected into the intramuscular or subcutaneous space [5,6,7]. The implant is formed via phase inversion, during which the water-miscible solvent diffuses from the ISFI into the aqueous injection site, leaving the precipitated polymer to trap the drug in a solid matrix [8,9]. Phase inversion kinetics have a strong effect on drug burst release (within the first 24 h) and can be tuned by changing system components, such as drug physicochemical properties or solvent, that alter the exchange between the aqueous environment and solvent [10]. The subsequent release rate is dictated by drug diffusion through the polymer matrix, and finally polymer degradation via hydrolysis [11]. Polymer hydrolysis rate is influenced by backbone structure, MW, crystallinity, hydrophilicity, microstructure, and material processing [12,13]. As a co-polymer, PLGA degradation can be fine-tuned by varying the ratio of lactic acid:glycolic acid, MW, molecular structure (linear vs. branched) and end group [14,15]. Previously, we showed that in vitro release of small-molecule drugs, MK-2048, and dolutegravir, from ISFI formulations can be tuned by altering the ratio of PLGA to NMP in the ISFI formulation [16]. The degradation byproducts of PLGA (lactic acid and glycolic acid) are non-toxic and known to be cleared through normal metabolic pathways, eliminating the need for surgical removal of the ISFI once all of the drug has been released [17].

ISFIs are an ideal drug delivery system for various applications, especially for those requiring frequent dosing regimens, because they are simple to manufacture, stable for months at room temperature, facile to administer, and can be removed to terminate treatment if required [16]. Although ISFIs are a ternary system, it is crucial to consider the impact that each formulation component has on drug release kinetics to achieve optimal therapeutic effects. Formulation development typically requires performing multiple studies using a wide range of polymer–solvent systems until the desired drug release is achieved, which is time-consuming and costly [18,19]. Factors that affect in vitro drug release from ISFIs include drug loading, drug and polymer properties, polymer/solvent ratio, ISFI manufacturing process, type of release media used, and injection volume of ISFI [20]. Formulation work can be improved by rationally designing drug release studies based on known physicochemical properties of drugs. Two physicochemical properties, pKa, a measure of acidity, and logP, a measure of hydrophilicity, are particularly important because the degradation of PLGA has been shown to occur rapidly in acidic environments and because the drug’s affinity for the aqueous environment dictates whether it will interact more with the solvent or the polymer [21,22]. Here, we performed a systematic in vitro drug release study over 90 days using antiviral and chemotherapeutic drugs with a wide range of logP and pKa values with a constant formulation of 50 mg/g API in 1:2 *w*/*w* PLGA:NMP in order to study the effects of drug physicochemical properties on in vitro drug release kinetics and PLGA degradation. The future goal of this study is to build a predictive model for drug release from PLGA ISFIs at relevant biological conditions. PLGA was chosen as the polymer in these ISFI formulations because it is biodegradable, tunable, and has been previously used in Food and Drug Administration (FDA)-approved ISFI systems. The long-term benefits of this study are three-fold: (1) the development of a predictive drug-release model will greatly benefit researchers in academia and industry due to the rapid development of novel small-molecule drugs [23], (2) the development of a long-acting implant mitigates the need for daily dosing and improves drug compliance, and (3) the long shelf-life of ISFI formulations when stored at room temperature will allow for distribution of therapies to countries where access to cold chain storage and distribution are limited [16].

## 2. Materials and Methods

### 2.1. Materials

50:50 Poly(DL-lactide-*co*-glycolide) ester terminated (PLGA) was purchased from LACTEL (Birmingham, AL; Cat. No. B6010-1P, Lot# A17-142, weight average MW, 27.2 kDa, intrinsic viscosity (i.v.) 0.38, polydispersity index (PDI) 1.81). *N*-methyl-2-pyrrolidone (NMP, (USP)) was received from ASHLAND (Wilmington, DE, USA, Product Code 830697, 100%NMP). Dolutegravir base (DTG), rilpivirine base (RPV), darunavir base (DRV), ritonavir base (RTV), etravirine base (ETV), efavirenz base (EFV), raltegravir base (RAL) and idarubicin hydrochloride salt (IDA) were purchased from Selleckchem (Houston, TX, USA). Gemcitabine base (GEM) was purchased from LC Laboratories (Woburn, MA, USA). Lamivudine base (3TC) was purchased from Fisher Scientific (Pittsburgh, PA, USA). Zidovudine base (ZDV) was purchased from MedChemExpress (Monmouth Junction, NJ, USA). 5-fluorouracil base (5FU), phosphate-buffered saline (0.01 M PBS, pH 7.4), solutol-HS, HPLC grade acetonitrile (ACN), and water were purchased from Sigma Aldrich (St. Louis, MO, USA).

### 2.2. Methods

#### 2.2.1. High-Performance Liquid Chromatography (HPLC)

Reverse-phase HPLC analyses were developed and validated with a Thermo Finnigan Surveyor HPLC (Thermo Finnigan, San Jośe, CA, USA) equipped with a Photodiode Array (PDA) Plus Detector, LC pump plus, and autosampler [16]. Sample analyses were carried out on an Intersil, ODS-3 column (4 µm, 4.6 Å ~ 150 nm (GL Sciences, Torrance, CA, USA) stationary phase maintained at 40 °C, with a flow rate of 1.0 mL/min with a 25 µL sample injection. A mobile phase of H_2_O:ACN 95:5 *v*/*v* and 0.1% trifluoroacetic acid (TFA) was used for the following active pharmaceutical ingredients (API) and read at (wavelength) with a total run time of 25 min: DTG (265 nm), RPV (265 nm), RTV (240 nm), ETV (240 nm), EFV (240 nm), DRV (265 nm), RAL (254 nm), ZDV (265 nm), IDA (254 nm). A gradient method was utilized to achieve separation (0–20 min: 5%–100% acetonitrile; 20–22 min: 100% acetonitrile; 23–25 min: 5% acetonitrile) [16]. A mobile phase of H_2_O:ACN 95:5 *v*/*v* and 0.1% trifluoroacetic acid (TFA) was used for the following drugs and read at (wavelength) with a total run time of (minutes): 5FU (265 nm, 10 min), GEM (280 nm, 20 min), 3TC (280 nm, 20 min).

#### 2.2.2. Preparation of ISFI Formulations

PLGA was mixed with NMP at a 1:2 weight ratio (*w*/*w*) and allowed to mix continuously at room temperature to fully dissolve PLGA and form a homogeneous placebo formulation [16]. All APIs were added at a constant concentration of 50 mg/g to the PLGA/NMP placebo formulation and allowed to mix at 37 °C overnight to yield a homogenous drug-loaded formulation. The homogeneity of drug-loaded ISFI formulations was assessed by collecting sample aliquots (1–2 mg, *n* = 4) from four different areas in the formulation and fully dissolving the samples in 1 mL of ACN overnight. For ZDV, 3TC and GEM, homogeneity samples were diluted in H_2_O at 100× after PLGA completely dissolved in ACN (2 h). Samples were filtered using a 0.2 µm nylon filter and then drug concentration was analyzed by HPLC analysis as described above. A formulation was considered homogeneous when the average concentration in all four aliquots had a standard deviation of ≤5%. A summary of drug concentrations and total drug per ISFI is included in Appendix A.

#### 2.2.3. Differential Scanning Calorimetry (DSC)

The DSC analysis of pure PLGA, pure drugs (DTG, RPV, RTV, ETV, EFV, DRV, RAL, GEM, 3TC, 5FU, IDA, and ZDV), placebo formulation (1:2 *w*/*w* PLGA:NMP), and drug-loaded ISFIs were carried out using a differential calorimeter (TA Q200, TA Instruments, New Castle, DE, USA) [24]. Pure drugs and pure PLGA were used as received. Before DSC analysis, the decomposition onset temperature (T_d_) of pure drugs was measured by thermogravimetric analysis (TGA) on a Thermogravimetric Analyzer (TA Q5000, TA Instruments, New Castle, DE, USA) [25]. The plots of wt% vs. temperature °C were used to calculate the values of T_d_ (the temperature at 5% weight loss). Pure drugs were heated from ambient temperature to 600 °C at a heating rate of 10 °C/min. ISFIs were prepared by injecting a solution of drug-loaded formulation (30 µL) into PBS (pH: 7.4, 37 °C), and incubating for 24 h. ISFIs were subsequently collected, air-dried for 48 h, and then lyophilized for 24 h (Labconco Corporation, Kansas City, MO, USA) before DSC analysis. Samples were weighed (1–10 mg), hermetically sealed in an aluminum pan, and placed in the differential scanning calorimeter. The samples were subsequently heated from 25 °C to a defined temperature below the T_d_, at a heating rate of 5 °C/min. The thermograms were used to determine the glass transition temperature (T_g_) for PLGA and the melting temperature peak (T_m_) for each API.

#### 2.2.4. In Vitro Cumulative Drug Release

In vitro drug release studies were carried out with formulations containing DTG, RPV, RTV, ETV, EFV, DRV, RAL, GEM, 3TC, 5FU, IDA, and ZDV [16]. Drug release kinetics from ISFI formulations were evaluated by injecting 30 µL of each ISFI solution (30 ± 5 mg, *n* = 4) into 200 mL of release medium (0.01 M PBS pH 7.4 with 2% solutol) and incubating the resulting implant under sink conditions at 37 °C for 90 days. Sample aliquots of 1 mL were collected at pre-determined timepoints and were replaced with fresh release medium. To maintain sink conditions and prevent bacterial growth, the release medium was completely removed and replaced with fresh release medium (200 mL) every week for the first 4 weeks and biweekly thereafter. Media were changed immediately after sampling to capture all of the drug release from the ISFIs. The drug concentration in the release medium was determined by HPLC using the methods described above. Cumulative drug release was calculated by HPLC analysis and was normalized by the total mass of drug in the implant. All experiments were performed in triplicate.

#### 2.2.5. Quantification of L-Lactic Acid (Lactic Acid Assay)

To quantify PLGA degradation over time, L-Lactic acid concentration in the release medium over a period of 90 days was quantified using a lactic acid assay kit (Sigma Aldrich-MAK064, St. Louis, MO, USA). To determine the correlation between PLGA degradation and in vitro drug release kinetics, the cumulative lactic acid release was calculated from the same sample aliquots collected to determine cumulative in vitro drug release by HPLC [26]. For this experiment, ISFIs were prepared by injecting a solution of drug-loaded formulation (30 µL) into PBS (pH = 7.4, 37 °C), and incubating for 24 h. The implants were collected, air-dried for 48 h followed by lyophilization for 24 h, and weighed. The dried implants were subsequently incubated in 20 mL of 5 M NaOH for 24 h at 37 °C to accelerate the hydrolysis of PLGA into its monomers and achieve complete hydrolysis to determine the total LA in PLGA used to prepare all ISFI formulations. The theoretical amount of lactic acid was calculated using the following equation: $\mathrm{Total}\;\mathrm{LA}\;\left(\mathrm{mg}\right)=\mathrm{PLGA}\;\mathrm{in}\;\mathrm{ISFI}\;\left(\mathrm{mg}\right)\times \frac{1}{2}\times \mathrm{Mass}\;\mathrm{of}\;\mathrm{depot};$ where 1 g of drug-loaded ISFIs have 317 mg PLGA and placebo ISFIs have 333 mg PLGA; and PLGA ½ corresponds to 50% LA in PLGA (50:50 LA/GA).

#### 2.2.6. Gel Permeation Chromatography (GPC)

Gel Permeation Chromatography (GPC) of placebo and drug-loaded ISFIs was performed on a EcoSEC Elite HLC-8420 GPC (Tosoh Biosciences, San Francisco, CA, USA) [27]. The GPC is also equipped with a Tosoh Lens 3 Multi-Angle Laser Light Scattering (MALS) detector (Tosoh Biosciences, San Francisco, CA, USA). The stationary phase used for the analysis was a TSKgel GMH-M column (7.8 mm × 30 cm with a pore size of 5 µm) (Millipore Sigma, St. Louis, MO, USA) maintained at 40 °C. Tetrahydrofuran (THF) was used as the mobile phase at a flow rate of 0.5 mL/min. MW and polydispersity index (PDI) were reported relative to polystyrene standards. ISFIs were obtained by injecting each drug-loaded formulation 30 µL (30 ± 5 mg, *n* = 4) into 200 mL of release medium (0.01 M PBS pH 7.4 with 2% solutol) and incubating under sink conditions at 37 °C. To maintain sink conditions, the release medium was completely removed and replaced with fresh medium (200 mL) every week for the first 4 weeks and biweekly thereafter over 90 days. On days 3, 30, 60, and 90, ISFIs (*n* = 1) for each drug loaded formulation) were collected, air dried at −20 °C for 48 h, and lyophilized for 24 h (SP VirTis Advantage XL-70, Warminster, PA, USA). ISFI samples (1 mg) were dissolved in THF (1 mL) and filtered through a 0.2 µm filter before GPC analysis using EcoSEC Elite^®^ (Tosoh Biosciences, San Francisco, CA, USA).

#### 2.2.7. Scanning Electron Microscopy (SEM) Imaging and Analysis

The microstructures of ISFIs were analyzed using SEM imaging [16]. Drug-loaded formulations (30 µL) were individually injected into 200 mL of release medium (0.1 M PBS with 2% solutol, pH 7.4) and incubated for 7 days at 37 °C. ISFIs were subsequently collected, flash-frozen using liquid nitrogen, and lyophilized for 24 h (SP VirTis Advantage XL-70, Warminster, PA, USA). The lyophilized samples were mounted on an aluminum stub using carbon tape, and sputter-coated with 5 nm of gold–palladium alloy (60:40) (Hummer X Sputter Coater, Anatech USA, Union City, CA, USA). The coated samples were then imaged using a Zeiss Supra 25 field emission scanning electron microscope with an acceleration voltage of 5 kV, 30 µm aperture, and an average working distance of 10 mm (Carl Zeiss Microscopy, LLC, Thornwood, NY, USA).

## 3. Results

The goal of this work was to study the effects of physicochemical properties of various drugs on PLGA degradation when used in ISFI formulations and on drug release kinetics in vitro. We selected the 1:2 PLGA:NMP formulation due to its use in the FDA-approved 7.5 mg Eligard^®^ ISFI system and due to its ability to sustain the release of small molecule drugs as shown in our previous work [16]. We also selected twelve small-molecule drugs based on two known physicochemical properties; the acid-base dissociation constant (pKa) and the lipophilicity of drugs (logP). The drugs were categorized into four groups: hydrophobic–acidic (rilpivirine (RPV), ritonavir (RTV), etravirine (ETV)), hydrophobic–basic (efavirenz (EFV), darunavir (DRV), dolutegravir (DTG)), hydrophilic–acidic (gemcitabine (GEM), lamivudine (3TC), raltegravir (RAL)), and hydrophilic–basic (idarubicin (IDA), zidovudine (ZDV), 5-fluorouracil (5-FU)) (Table 1). These drugs were selected for this study based on their indication and mechanism of action and included anti-HIV and chemotherapeutic drugs. Among the antiretroviral drugs, we studied non-nucleotide reverse transcriptase inhibitors (NNRTIs) RPV, ETV, and EFV, nucleotide reverse transcriptase inhibitors (NRTIs) ZDV and 3TC, integrase strand transfer inhibitors (ISTIs) DTG and RAL, and the protease inhibitors DRV and RTV. The chemotherapeutic drugs studied were GEM and 5-FU, which are anti-metabolites, and IDA, which is an anthracycline.

To study the quantitative and qualitative effects of each drug on PLGA degradation and drug release kinetics in vitro, we performed assays on drug release (HPLC), lactic acid release (fluorescence), PLGA MW (GPC), drug physical state (DSC), and depot morphology (SEM) as outlined in Figure 1. For each study, formulation parameters including PLGA (50:50 LA/GA, MW 27 kDa), PLGA:NMP ratio (1:2 *w*/*w*), drug concentration (50 mg/mg), and depot size (30 mg) were kept constant to isolate the effect of each drug on PLGA degradation and in vitro release kinetics. The first study, depicted in Figure 1B, was designed to measure drug release from the depots and lactic acid release as a result of PLGA hydrolysis. Sample aliquots (1 mL) of media were collected through 90 days and replaced with fresh media; 500 µL was used for drug release analysis and 500 µL was used for lactic acid release analysis. The media was fully replaced as shown in Figure 1B to maintain sink conditions. The second study, depicted in Figure 1C, was designed to measure the MW of PLGA in the ISFIs and qualitatively observe the drug’s physical state and depot morphology. ISFIs were generated using the aforementioned method and collected on days 1, 7, 30, 60, and 90 post-injection into PBS at 37 °C. ISFIs were subsequently lyophilized for 24–48 h before analysis by each assay.

### 3.1. In Vitro Release Kinetics and Lactic Acid Release

#### 3.1.1. Drug Burst Release 1 Day Post-ISFI Injection

When designing an LA-injectable formulation, it is important to control the initial drug burst release within the first 24 h to avoid potential toxicity or side effects as a result of large drug concentration. In this study, we investigated the effect of drug physicochemical properties on the initial burst release within the first 24 h post-ISFI injection into the release media in vitro. Results showed that hydrophobic drugs exhibited low burst release (1.7–18.9%) and had a direct correlation between drug logP and the initial burst where burst release decreased as logP increased (Figure 2A). We hypothesize that as the hydrophobicity of the drug increases, its affinity for the hydrophobic polymer increases and its affinity for the hydrophilic solvent and aqueous medium decreases, reducing the 24 h burst release. On the other hand, hydrophilic drugs exhibited high burst release within the first 24 h (22.8–68.3%) due to their high affinity for NMP and the aqueous release media leading to rapid diffusion during the phase inversion upon injection into the media (Figure 2B). Unlike hydrophobic drugs, a trend between logP and burst release was not observed for hydrophilic drugs. Notably, IDA had the highest burst release at 68.3% likely due to its higher aqueous solubility in its salt form [28]. For this reason, additional rate-controlling excipients should be added to ISFIs with hydrophilic drugs to reduce the initial burst release [29,30].

#### 3.1.2. Overall Drug Release Kinetics

In vitro drug release kinetics were assessed over 90 days for each ISFI formulation (Figure 3). A typical drug release profile of PLGA ISFIs consists of three phases: (1) burst release within the first 24 h, (2) diffusion-mediated release, and (3) polymer degradation-mediated release [31]. Within the second phase of release kinetics (day 2–30) where drug release is mainly governed by drug diffusion before initiation of polymer bulk degradation, there was a clear correlation between drug logP and percent drug release for hydrophobic drugs regardless of their pKa (Figure 3A,B). In the first 30 days of incubation, within the class of hydrophobic drugs with basic pKa, drug release was inversely related to logP, where DRV with the lowest logP of 1.8 exhibited the slowest release kinetics (11.1 µg/day; 41.5% at day 30) compared with DTG (logP 2.2; 15.0 µg/day; 48.0% at day 30) and EFV (logP 4.6; 29.3 µg/day; 62.5% at day 30) (Figure 3A). On the other hand, within the class of hydrophobic drugs with acidic pKa, drug release was directly related to logP and ETV with the highest logP of 5.54 exhibiting slower release kinetics (1.66 µg/day; 4.8% at day 30) compared with RPV (logP 4.86; 9.03 µg/day; 17.7% at day 30) and RTV (logP 3.9; 18.2 µg/day; 44.4% at day 30) (Figure 3B). Unlike hydrophobic drugs, hydrophilic drugs except for RAL exhibited fast diffusion and complete release within the first 30 days (Figure 3C,D). RAL exhibited very slow diffusion within the first 30 days followed by a sharp increase in release kinetics driven by PLGA bulk degradation (Figure 3D). The slow diffusion of RAL from the ISFI could be due to RAL’s higher affinity for PLGA via multiple sites of hydrogen-bonding interactions. Siegel et al. also found that water solubility, which is related to the drug’s logP, leads to faster drug release; however, there were exceptions to this, demonstrating that there could be other factors at play [32]. Concerning hydrophobic drugs, most drugs exhibited overall slower release kinetics that would last beyond 90 days (26.7–80.6% at day 90 for ETV, RPV, RTV, and DTG). Amongst hydrophobic drugs, DRV (pKa 11.4) and EFV (pKa 10.2) exhibited relatively faster release rates and achieved complete release by day 60 (Figure 3A). The faster release kinetics of DRV and EFV could be attributed to their strong basic nature (pKa > 10), which can promote ester hydrolysis of PLGA [33,34]. In addition, when formulated in an ISFI, DRV and EFV exhibited mainly an amorphous state in the depot as shown by SEM imaging. GPC analysis showed that both DRV and EFV resulted in a greater percent decrease in PLGA MW at day 30 (11.8% for DRV and 20.4% for EFV) compared with the percent decrease of PLGA MW for placebo ISFI on day 30 (6.7%). Additionally, DRV which has the lowest logP amongst the hydrophobic drugs exhibited the highest burst release in the first 24 h (Figure 2A). 

The onset of degradation mediated drug release for the strongest basic drugs EFV and DRV matched the onset of PLGA bulk degradation, which is known to be initiated at day 30 (Figure 4A). Surprisingly, only RPV had a significant effect on PLGA degradation, determined by quantifying LA release over time, resulting in an early onset of bulk degradation at day 21 compared with placebo ISFI where the onset of PLGA degradation was at day 28 post-incubation at 37 °C (Figure 4B). This effect was also reflected by a statistically significant increase of lactic acid release on day 30 for RPV ISFI compared to placebo ISFI (Figure 5A), which correlates with the increase in RPV release on day 30 (Figure 4B). Furthermore, GPC analysis of RPV-loaded ISFIs also demonstrated a greater percent PLGA MW decrease at day 30 (15.85%) compared with placebo ISFI (6.64%) (Figure 6B).

For hydrophilic drugs, except RAL, given that complete drug release was achieved within the first 30 days, the effects of drug properties (logP, pKa) on PLGA degradation (MW, lactic acid release) can be assessed when drugs were still present in the ISFI (days 1–28) and after drugs are completely released from the ISFI (days 30–90). The effect of logP and pKa on PLGA degradation was most evident for 3TC (logP: −0.49, pKa: 3.4) and GEM (logP: −1.4, pKa: 4.3), which exhibited a faster onset of PLGA bulk hydrolysis at day 14 and 21, respectively, compared with day 30 with placebo ISFIs (Figure 4C,D). This effect was corroborated with higher lactic acid release at day 30 for GEM and 3TC (10.3% and 11.0% LA respectively) compared with placebo ISFI (6.7% LA). However, this effect was reversed after 45 days of incubation, where placebo ISFI exhibited faster degradation (35.9% on day 90) compared with 3TC and GEM (18.4% and 28.3% on day 90 respectively). On the other hand, RAL ISFIs exhibited a significantly lower % lactic acid release at day 30 (1.1%, *p* = 0.0002) and day 90 (6.0%, *p* < 0.0001) compared with placebo ISFIs (6.7% and 35.9% at day 30 and 90 respectively (Figure 5C,D). Overall, ISFIs prepared with all twelve drugs tested exhibited lower percent lactic acid release at day 90 compared with placebo ISFIs (Figure 5B,D). In vitro drug release kinetics and lactic acid release are also compared for each drug in Appendix A and normalized to 100% at day 90 in Appendix A.

### 3.2. Gel Permeation Chromatography (GPC) Analysis

The percent decrease in PLGA MW in both placebo and drug-loaded ISFIs was monitored using GPC analysis over 90 days (Appendix A). GPC analysis showed significant degradation over the 90-day in vitro release studies in PBS (pH 7.4 with 2% solutol) at 37 °C for both placebo and drug-loaded ISFIs (Figure 6). ISFIs formulated with hydrophobic drugs (acidic and basic) exhibited a similar percent MW decrease compared to placebo ISFIs (Figure 6A,B). On the other hand, ISFIs formulated with hydrophilic drugs showed a greater percent MW decrease, particularly at days 30 and 60 post-injection, compared with placebo ISFIs (Figure 6C,D). The faster PLGA degradation kinetics in presence of hydrophilic drugs correlate with faster release kinetics observed with IDA, 5FU, GEM, and 3TC (Figure 3C,D). The effect of PLGA MW decreasing as a result of pH could not be determined due to weekly media change in the first 30 days that was required to maintain sink conditions for all drugs.

### 3.3. Scanning Electron Microscopy (SEM) Analysis

Drug release behavior from ISFIs is greatly influenced by the implant’s microstructure. Changes in microstructure can be attributed to polymer, solvent, drug properties, and polymer degradation rate [8,9,10]. SEM imaging was used to investigate the effect of drug physicochemical properties (MW, logP, and pKa) on the microstructure of the PLGA implant and explain drug release behavior. Drugs can be identified in each image by rose-like shapes, rods, or crystals. As shown in Figure 7, all implants elicited porous structures likely resulting from drugs’ high solubility in NMP (>50 mg/g), which can result in a drug being released as NMP effluxes from the implant [8]. Specifically, in Figure 7B, DRV ISFIs had the largest pore sizes amongst implants loaded with hydrophobic drugs, which directly correlated with DRV exhibiting the largest burst release (Figure 1). Additionally, SEM images of DTG ISFIs (Figure 7B) showed that DTG was mainly molecularly dispersed within the implant with some larger pores, which could also contribute to its high burst release compared with other implants loaded with hydrophobic drugs. Unlike ISFIs loaded with hydrophobic drugs, ISFIs loaded with hydrophilic drugs (Figure 7C,D) elicit a large hollow center core as a result of the high drug burst release and fast phase inversion. This hollow core can be seen with GEM, 3TC, ZDV, and 5FU ISFIs. This trend in microstructure was not observed with RAL ISFIs or IDA ISFIs. RAL ISFIs microstructure was similar to ISFIs loaded with hydrophobic drugs and exhibited a porous structure and observable crystalline drug particles, which can explain its low burst release compared with other hydrophilic drugs, despite its negative logP value. On the other hand, IDA ISFIs elicited very large porous structures and apparent fractures within the implant microstructure, which is likely attributed to IDA’s relatively high burst and cumulative release compared with the other hydrophilic drugs.

## 4. Discussion

Overall, the present study demonstrated how physicochemical properties of drugs (logP and pKa) can be used to rationally design sustained drug release formulations. In PLGA-based formulations, hydrophobic drugs can reside in the polymer matrix for much longer than hydrophilic drugs due to their affinity for the polymer. The trend observed in burst release versus logP can be used to predict burst release for other hydrophobic drugs, for which burst release may be desired or undesired. Conversely, hydrophilic drugs release much more quickly than hydrophobic drugs, particularly on the first day while the depot is still solidifying, due to their affinity for the aqueous environment. The effect of drug pKa was predicted to have a more direct impact on the degradation of PLGA and drug release kinetics, given that changes in local pH can increase or decrease ester hydrolysis; however, these trends were difficult to observe. This could be attributed to the weekly media change during the first 30 days of in vitro release studies to maintain sink conditions, which would minimize the amount of time hydrophilic drugs have to come in contact with PLGA. Finally, drug crystallinity in the polymer matrix can dictate release kinetics; as shown in our SEM data, hydrophobic drugs tended to remain crystalline while hydrophilic drugs were mostly amorphous and molecularly dispersed within the ISFI. Taken together, these findings will inform and aid formulation development efforts for small molecule drugs in in situ forming implants and provide a framework for understanding how drugs with differing physicochemical properties can affect PLGA degradation. Future work can analyze the effects of other ISFI parameters such as drug loading, ISFI injection volume, polymer-to-solvent ratio, and media type; finally, these effects can also be studied in small animal models to understand the effects of injection site biology on ISFI formation and degradation.

## Figures and Tables

**Figure 1 pharmaceutics-14-01188-f001:**
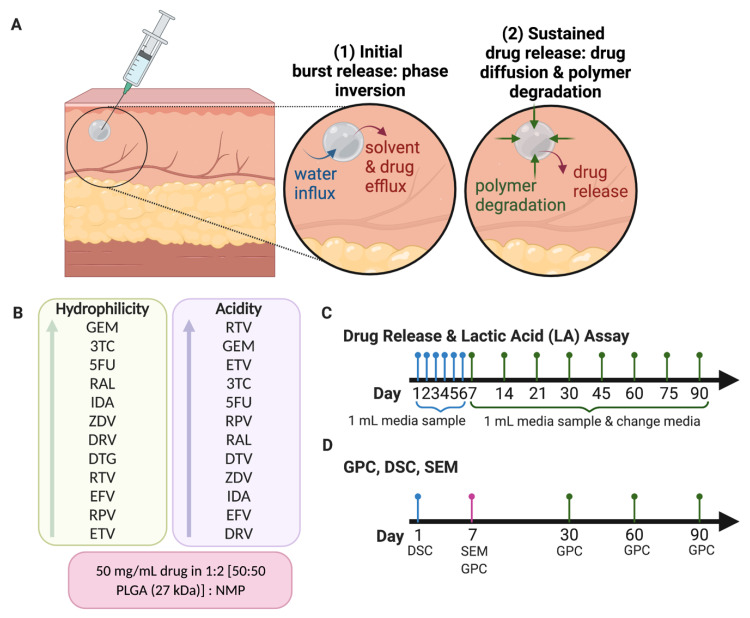
Study outline to determine the effects of drug physicochemical properties on ISFI degradation and drug release kinetics in vitro. Figure created with BioRender.com, accessed on 6 May 2020. (**A**) Schematic of ISFI formation and degradation. (**B**) Twelve drugs used in the study organized by logP and pKa. All formulations were prepared with a constant drug loading of 50 mg/g in 1:2 *w*/*w* PLGA:NMP (PLGA 50:50 LA/GA, 27 kDa), and depot sizes were maintained at 30 mg in all studies. (**C**) ISFIs (*n* = 4) were incubated in 200 mL of release media (PBS with 2% solutol) at 37 °C and sample aliquots (1 mL) were collected at predetermined time points and analyzed to quantify drug and lactic acid release over 90 days. (**D**) Lyophilized ISFIs (*n* = 1) were analyzed by GPC, DSC, and SEM to determine PLGA MW, PLGA and drug physical state, and ISFI microstructure, respectively.

**Figure 2 pharmaceutics-14-01188-f002:**
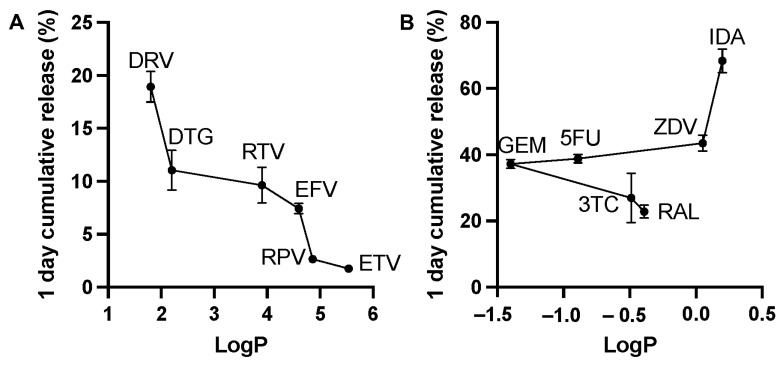
Initial in vitro burst release quantified 24 h post ISFI incubation in PBS (pH 7.4 with 2% solutol) at 37 °C plotted relative to drug logP. (**A**) burst release of 50 mg/g DRV (logP 1.8), DTG (logP 2.2), RTV (logP 3.9), EFV (logP 4.6), RPV (logP 4.8) and ETV (logP 5.5) ISFIs; (**B**) burst release of 50 mg/g GEM (logP −1.4), 5FU (logP −0.89), 3TC (logP −0.49), RAL (logP −0.39), ZDV (logP 0.05) and IDA (logP 0.2) ISFIs.

**Figure 3 pharmaceutics-14-01188-f003:**
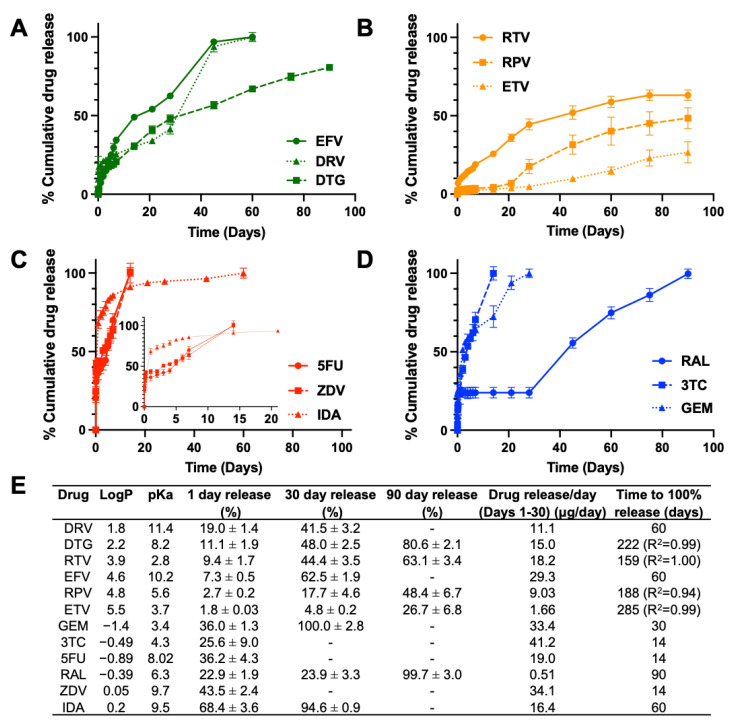
Cumulative percentage in vitro drug release kinetics from drug-loaded ISFIs (50 mg/g drug in 1:2 PLGA:NMP) over a course of 90 days for (**A**) hydrophobic–basic drugs, (**B**) hydrophobic–acidic drugs, (**C**) hydrophilic–basic drugs (inlay for Days 0–21), and (**D**) hydrophilic–acidic drugs. In vitro drug release studies were performed in 0.01 M PBS and 2% solutol at pH 7.4 and 37 °C under sink conditions. (**E**) Summary table for in vitro drug release kinetics.

**Figure 4 pharmaceutics-14-01188-f004:**
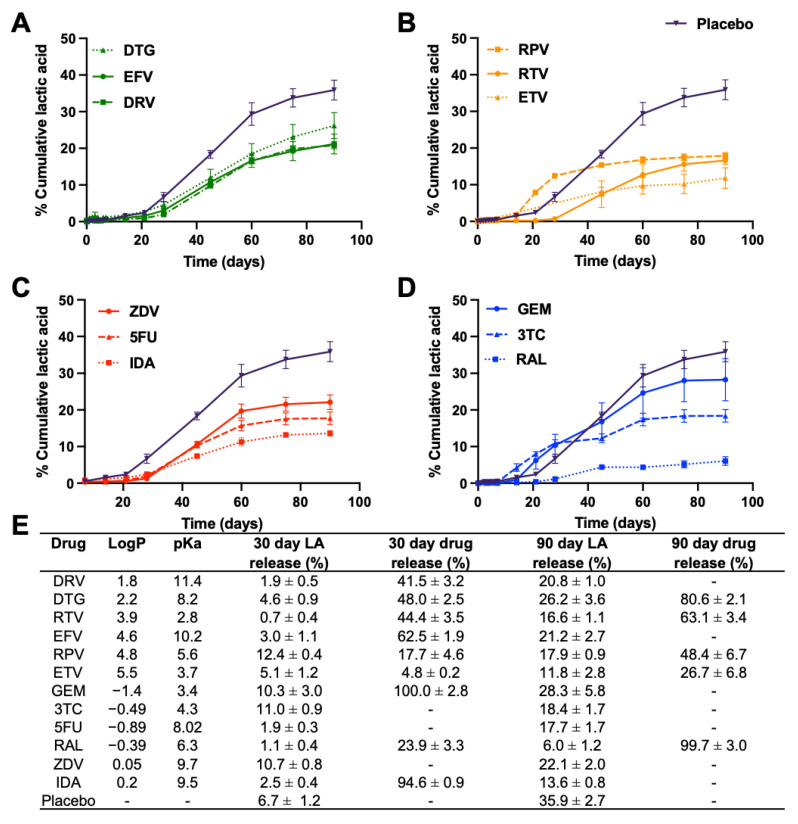
Cumulative percentage in vitro LA release kinetics from drug-loaded ISFIs (50 mg/g drug in 1:2 PLGA:NMP) over a course of 90 days for (**A**) hydrophobic–basic drugs, (**B**) hydrophobic–acidic drugs, (**C**) hydrophilic–basic drugs, and (**D**) hydrophilic–acidic drugs. In vitro LA release studies were performed in 0.01 M PBS and 2% solutol at pH 7.4 and 37 °C under sink conditions. (**E**) Summary table for in vitro drug release kinetics.

**Figure 5 pharmaceutics-14-01188-f005:**
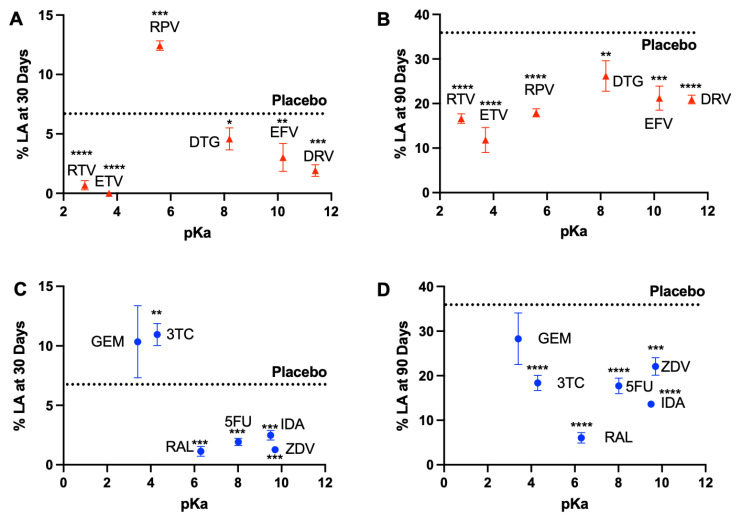
Cumulative percent in vitro release of lactic acid from drug-loaded ISFIs (50 mg/g drug in 1:2 *w*/*w* PLGA:NMP) as a function of drug pKa at day 30 and day 90 post-injection into release media (PBS, pH 7.2 with 2% solutol) at 37 °C relative to placebo ISFIs (**A**) In vitro lactic acid release kinetics of hydrophobic drugs at day 30 post-injection. RTV *p* < 0.0001; ETV *p* < 0.0001; RPV *p* = 0.0001; DTG *p* = 0.0345; EFV *p* = 0.0053; DRV *p* = 0.0004. (**B**) In vitro lactic acid release kinetics of hydrophobic drugs at day 90 post-injection. RTV *p* < 0.0001; ETV *p* < 0.0001; RPV *p* < 0.0001; DTG *p* = 0.0044; EFV *p* = 0.0003; DRV *p* < 0.0001. (**C**) In vitro lactic acid release kinetics of hydrophilic drugs at day 30 post-injection. GEM *p* = 0.0674; 3TC *p* = 0.0016; RAL *p* = 0.0002; 5FU *p* = 0.0003; IDA *p* = 0.0007; ZDV *p* = 0.0002. (**D**) In vitro lactic acid release kinetics of hydrophilic drugs at day 30 post-injection. GEM *p* = 0.0540; 3TC *p* < 0.0001; RAL *p* < 0.0001; 5FU *p* < 0.0001; IDA *p* < 0.0001; ZDV *p* = 0.0002. (*: *p* < 0.05, **: *p* < 0.01, ***: *p* < 0.001, ****: *p* < 0.0001.)

**Figure 6 pharmaceutics-14-01188-f006:**
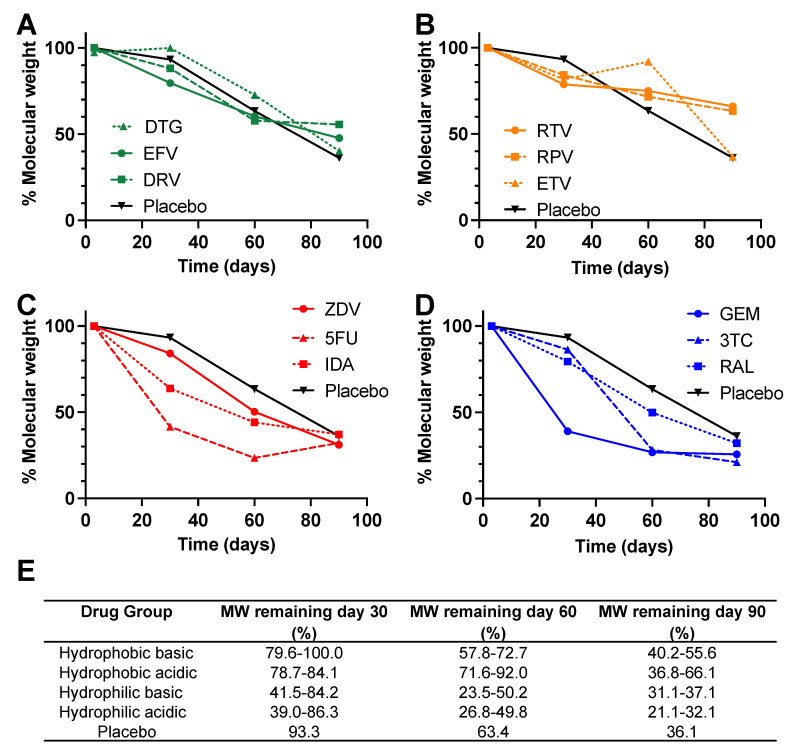
Percent PLGA MW decrease of drug-loaded ISFIs (50 mg/g drug in 1:2 *w*/*w* PLGA:NMP) over 90 days for (**A**) hydrophobic–basic drugs, (**B**) hydrophobic–acidic drugs, (**C**) hydrophilic–basic drugs, and (**D**) hydrophilic–acidic drugs. (**E**) Shows the numerical values of % molecular weight remaining at days 30, 60, and 90. The initial MW of PLGA (100% MW; time 0) for each ISFI was determined by analyzing depots post 3 days incubation in PBS at 37 °C to allow full removal of solvent from the ISFI. The remainder of PLGA MW data were normalized as a percent of the initial PLGA MW quantified at time 0.

**Figure 7 pharmaceutics-14-01188-f007:**
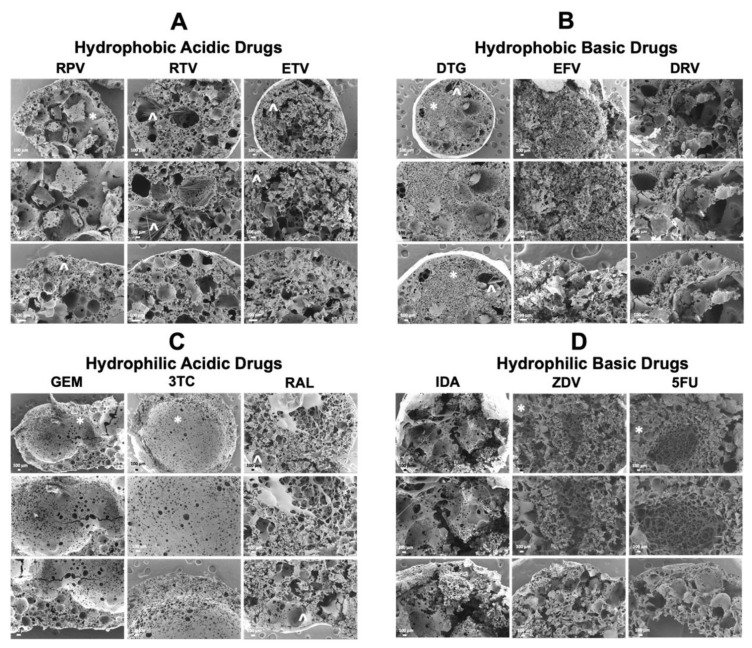
SEM cross-section images of drug-loaded ISFIs. (**A**) Implants with hydrophobic–acidic drugs (RPV, RTV, and ETV). (**B**) Implants with hydrophobic–basic drugs (DTG, EFV, and DRV). (**C**) Implants with hydrophilic–acidic drugs (GEM, 3TC, and RAL). (**D**) Implants with hydrophilic–basic drugs (IDA, ZDV, and 5FU). Each column within A, B, C, and D matrix represents increasing magnification (60×, 100×, and 200×). Scale bar = 100 µm. (*) denotes polymer and (^) denotes drug.

**Table 1 pharmaceutics-14-01188-t001:** Physicochemical properties of drugs tested in ISFIs.

Drug	Chemical Structure	MW(g/mol)	LogP *	pKa *	Class
Rilpivirine(RPV)	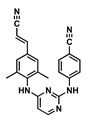	366.42	4.86	5.6	ARV-NNRTI
Ritonavir(RTV)	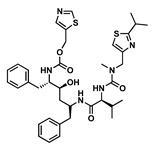	720.95	3.90	2.8	Booster
Etravirine(ETV)	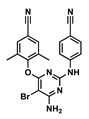	435.28	5.54	3.7	ARV-NNRTI
Efavirenz(EFV)	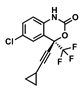	315.6	4.6	10.2	ARV-NNRTI
Darunavir(DRV)	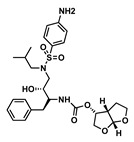	547.6	1.8	11.4	ARV-PI
Dolutegravir(DTG)	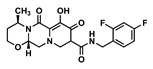	419.3	2.2	8.2	ARV-ISTI
Gemcitabine(GEM)	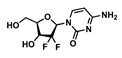	263.2	−1.4	3.6	Antimetabolite
Lamivudine(3TC)	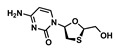	229.2	−0.49	4.8	ARV-NRTI
Raltegravir(RAL)	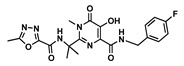	444.4	−0.39	6.3	ARV-ISTI
Idarubicin(IDA)	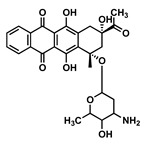	497.5	0.2	9.5	Anthracycline
Zidovudine(ZDV)	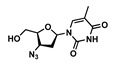	267.24	0.05	9.5	ARV-NRTI
5-Fluorouracil(5FU)	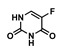	130.27	−0.89	8.02	Antimetabolite

* The acid base constant (pKa) and the log of drug partition coefficient (logP) were obtained from drugbank.ca (accessed on 21 July 2020). ARV—Antiretroviral. NNRTI—Non-nucleotide Reverse Transcriptase Inhibitors. NRTI—Nucleotide Reverse Transcriptase Inhibitors. ISTI—Integrase Strand Transfer Inhibitor. PI—Protease Inhibitor.

## Data Availability

The data presented in this study are within the article and Appendix A, or on request from the corresponding authors.

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
