# Peer review of "Effects of Drug Physicochemical Properties on In-Situ Forming Implant Polymer Degradation and Drug Release Kinetics"

_pharmaceutics, 2022, doi:10.3390/pharmaceutics14061188_

Round 1
Reviewer 1 Report
Referee Report
Title: Effects of drug physicochemical properties on in-situ forming implant polymer degradation and drug release kinetics
Manuscript ID: pharmaceutics-1722294
By Joiner et al
Submitted to Pharmaceutics (ISSN 1999-4923)
Comments
This work studied the effects of physicochemical properties of drug on an in-situ forming implant regarding polymer degradation and drug release kinetics. This work evaluated the in-vitro drug release of 12 drugs focusing on the lipophilicity of drugs and acid-base dissociation constant values from the in-situ forming implants. This work is well-written, comprehensive and thorough. I only have some minor comments:
- Introduction: Is it possible to provide a schematic diagram showing the functionality and mechanism of ISFIs in drug delivery?
- Materials: It would be useful to have a Table to provide details of the 12 drugs (DTG, RPV, RTV, ETV, EFV, DRV, RAL, GEM, 3TC, 5FU, IDA, and ZDV).
- Table 1: Order of the drugs in the table should follow the list in the text. That is, ZDV should be in the last row. Moreover, it is good to have more space between each row of the chemical structure (ZDV and IDA seems combined together in the table).
- Figure 1: Please be consistence to use the long or short form of the drugs. In this case, short form of the drugs should be used.
- Figure 2: Error bars of EFV, RPV and ETV are missing in 2(A), and Error bar of GEM is missing in 2(B).
- Figure 3(C): Is it possible to rescale the range in the x-axis from 0-100 days to 0-60 or 0-70 days? With such extended scale, we can view the curves clearer. The authors should note curves of ZDV and 5FU overlapped significantly and are difficult to view.
- Figure 5(B): The x-axis label should read “pKa” instead of “pka”.
- Error bars are missing in Figure 6. Figure 6(E) should be a separated Table with caption.
- Please provide ruler scales of SEM for Figure 7.
Author Response
We thank the reviewer for the valuable comments. We have addressed all your comments to the best of our ability in the revised manuscript and provided a point by point response to each comment you made (see attached response letter).

Reviewer 2 Report
The manuscript “Drug physicochemical properties affect in-situ forming implant polymer degradation and drug release” is designed well and interesting for the readers working on development of in-situ gel phenomenon-based drug products and also hydrogel formulations. However, it needs some edits before acceptable for publication.
- The research mainly focused on drug logP and pKa. But the in-vitro release of the drug also varied with drug loading, manufacturing process, release media concentrate, release studies process parameters and type of release retarded polymer uses. Discuss the effect of these parameters from the ISFIs.
- Hydrophilic drugs exhibit high initial burst and less sustained release due to their miscibility with the aqueous phase. But still, we could modify the release behavior with ER and IR mixing portions through osmotic, chronomodulated like delivery systems. Please explain?
- Introduction of the manuscript needs improvement. It lacks the why PLGA polymer ISFIs selected in the design.
- Materials – please specify the form (salt/base) of the drugs used.
- HPLC method – write the source of the method adopted or write the validation parameters if newly developed.
- The homogeneity of drug-loaded ISFI formulations was assessed by collecting sample aliquots (1-2 mg, n=4) from four different areas in the formulation and quantifying drug concentration in the samples by HPLC analysis. Elaborate how to prepare the samples for analysis (solvent used, dilution, etc).
- Write the reason for selection of 1:2 ratio of PLGA:NMP for the formulations in the results section.
- Line# 129, neat PLGA – change to pure PLGA.
- Write the type of method used for in vitro release studies.
- How did conform the recrystallization effect of the polymer over a period of release studies.
- Supplementary data not cited in the manuscript body.
Author Response

(The authors gave the same response as above.)

Reviewer 3 Report
In this manuscript, the effects of physicochemical properties of 12 drugs used in ISFI formulations on PLGA degradation and in vitro drug release kinetics were investigated. This research paper has certain scientific significance. But the Authors should address and clarify several points to make the manuscript suitable for publication, as follow:
- Line 22: Please define “PLGA”in the abstract. This is the first time this abbreviation appears in the manuscript.
- Line 151-153: “To maintain sink conditions, the release medium was completely removed and replaced with fresh release medium (200 mL) every week for the first 4 weeks and biweekly thereafter.”I don't quite understand switching to fresh release medium. Maybe authers just want to make the sink condition clean? However, will the change of release medium affect the release of drugs in this experiments? The authors need to explain this.
- Line257-259: “Results showed that hydrophobic drugs exhibited low burst release (1.7-18.9%) and had a direct correlation between drug logP and the initial burst where burst release decreased as logP increased (Figure 2A).” I would like the authors to add further analysis of this result.
- The authors should add error bars to the line chart in Figure 6.
- There istoo little analysis of the experimental results in the The author should consult more literature to make a reasonable analysis of the experimental results, rather than state the experimental results too much.
- Line 382-384: “Drug release behavior from ISFIs is greatly influenced by the implant’s microstructure. Changes in microstructure can be attributed to polymer, solvent, drug properties, and polymer degradation rate.”Authors should add a reference here.
- Line 387-389: “As shown in Figure 7, all implants elicit porous structures likely resulting from drugs’ high solubility in NMP (>50 mg/mL), which can result in drug being released as NMP effluxes from the implant.”Authors should add a reference here.
Authors should add references to Methods part as appropriate.
Author Response

(The authors gave the same response as above.)

Reviewer 4 Report
The purpose of this manuscript was to characterize the in-vitro drug release of 12 different small molecule drugs with differing logP and pKa values from ISFIs. The drug crystallinity in the polymer matrix can dictate release kinetics; as shown in the SEM data, hydrophobic drugs tended to remain crystalline while hydrophilic drugs were mostly amorphous and molecularly dispersed within the ISFI. Taken together, these findings will inform and aid formulation development efforts for small molecule drugs in in-situ forming implants and provide a framework for understanding how drugs with differing physicochemical properties can affect PLGA degradation. Although this manuscript is interesting and well-written, the polymer degradation and drug release kinetics should also be investigated in vivo by implanting the PLGA-drug in animals.
Author Response

(The authors gave the same response as above.)

Reviewer 5 Report
The authors have investigated the effect of drug physicochemical properties on in-situ forming implant (ISFI) polymer degradation and drug release kinetics. toxicological and functionalized formulations of carbon nanotubes (CTNs). The ISFI formulations were prepared and characterized physicochemically and evaluated for drug release studies. The topic of manuscript is interesting for publication, however, it needs significant revisions before acceptance. My suggestions are listed below:
Abstract: The quantitative information is not included in the abstract. Authors are advised to include some quantitative information in order to enhance the readability of the article.
Introduction: The rationale and objective of studies are not clear.
Sections 2.2.1. Please clearly indicate whether the HPLC method was developed or it is reported one. Also provide its validation status.
The prepared ISFI formulations were characterized for DSC only. Authors are advised to include more physicochemical parameters in order to validate the proper formation of formulations.
Table 1: Define the abbreviations in continuous sentence. Chemical structures are well-known for each drug. Please remove them.
Figure 7: The size bar is included. Kindly include it.
Kindly include the city and country of location for all the manufacturers of chemicals, instruments, and software.
Kindly include a suitable literature for each methodology.
Author Response

(The authors gave the same response as above.)

Reviewer 6 Report
The authors have made an interesting article describing the effects of drug physicochemical properties on in-situ forming implant polymer degradation and drug release kinetics but for further publication in the MDPI Journal of Pharmaceutics, some improvements should be made:
- The writing formation is not uniform done and not properly aligned in the left, it is not written with Justify, aligned right and left.
- The formulation “in an aluminium pan” from Line 140 should be replaced
- In Figure 2 why not in the representation for EFV, RPV, ETV is represented the standard deviation with error bar?
- Also in Figure 4 the standard deviation is not represented in Fig. 4 B for RPV, in Figure 4C for IDA and also in Figure 4D for RAL.
Author Response

(The authors gave the same response as above.)

Round 2
Reviewer 1 Report
I am satisfied with the modifications and additional contents added by the authors as per my comments. The presentation and quality of this revised manuscript are improved. I have no further questions.
Reviewer 2 Report
Manuscript modified as per the suggested edits.
Reviewer 4 Report
The authors have addressed my comments.
Reviewer 5 Report
The authors have addressed the previous concerns. The revised manuscript is suitable for publication in its present form.